# Association of Gastric Myoelectric Activity with Dietary Intakes, Substrate Utilization, and Energy Expenditure in Adults with Obesity

**DOI:** 10.3390/nu14194021

**Published:** 2022-09-28

**Authors:** Mahmoud M. A. Abulmeaty, Ghadeer S. Aljuraiban, Dara Aldisi, Batool Albaran, Zaid Aldossari, Thamer Alsager, Suhail Razak, Yara Almuhtadi, Eman El-Shorbagy, Mohamed Berika, Mohamed Al Zaben, Ali Almajwal

**Affiliations:** 1Community Health Sciences Department, College of Applied Medical Sciences, King Saud University, Riyadh 11433, Saudi Arabia; 2Obesity Management Unit, Medical Physiology Department, School of Medicine, Zagazig University, Zagazig 44519, Egypt; 3Rehabilitation Sciences Department, College of Applied Medical Sciences, King Saud University, Riyadh 11433, Saudi Arabia; 4Surgery Department, Sultan Bin Abdulaziz Humanitarian City, Riyadh 1564, Saudi Arabia

**Keywords:** gastric myoelectric activity, substrate utilization, energy expenditure, dietary intake, obesity

## Abstract

Obesity can modulate gastric myoelectric activity (GMA); however, the relationship of GMA with nutrient intakes and substrate utilization in adults with obesity is lacking. We examined the association of dietary intakes, energy expenditure, and substrate utilization with the GMA. Participants (*n* = 115, 18–60 y) were divided into healthy weight (HW, *n* = 24), overweight (OW, *n* = 29), obese (OB, *n* = 41) and morbidly obese (MO, *n* = 21). Two-day multi-pass 24 h recalls were conducted. The GMA was measured by multichannel electrogastrography (EGG) with water-load (WL) testing. Resting metabolic rate (RMR) and percentages of substrate utilization were measured by indirect calorimetry. In the HW, protein intake was directly correlated with average dominant frequency (ADF) and with WL volume, while in obese participants and the MO subgroup, WL volume correlated with carbohydrate intake. In participants with obesity, ADF was positively correlated with fiber intake. In participants with obesity and the OB subgroup, RMR was positively correlated with water-load volume (r = 0.39 and 0.37, *p* < 0.05). The ADF showed negative correlations with percent of fat utilization and positive correlations with percent of CHO utilization in non-obese groups. However, protein utilization showed inverse correlation in all obese groups. In conclusion, these distinctive associations suggest that certain dietary compositions and dieting regimens impact GMA patterns.

## 1. Introduction

Obesity is a complex health condition and a major contributor to the burden of chronic disease [1]. It is defined as an endocrine–metabolic and multifactorial disease involving the interaction of genetic, environmental, and hormonal factors, triggering adiposity in excess [2,3]. Individuals with obesity are at high risk of developing a variety of physical and mental comorbidities, such as cardiovascular diseases, diabetes, or musculoskeletal disorders. Moreover, obesity has been linked with dysfunctions of the autonomic nervous system (ANS) and functional gastrointestinal disorders (FGIDs) [4].

Research efforts for an effective treatment approach mainly focus on diet and exercise programs to determine the most effective recommendations for sustained body weight loss [5]. Diet intervention studies suggest spontaneous weight loss following low-fat diets and current data on reducing the carbohydrate-to-protein ratio of the diet shows promising outcomes [5]. In addition, resting metabolic rate (RMR) is affected by calories consumed in the diet [6]. Too much energy consumption seems to increase the resting metabolic rate while fasting and very-low-calorie dieting causes a lowering of RMR. Since the resting metabolic rate is the key component in daily energy expenditure, decreasing caloric restriction makes it challenging for the obese to lose or sustain the lost weight [6].

Gastric myoelectrical activity (GMA) is associated with the ANS and food’s ingested volume and composition. A frequency range between 2.03 and 4.06 cpm is regarded as normal gastric activity [7]. Electrogastrography (EGG) is a non-invasive procedure to record GMA through electrodes positioned on the surface of the abdomen [8]. Our previous work reported an association between GMA and gastric hormonal release [9]. Gastric slow waves are electrical impulses that spread along the stomach, generated by interstitial cells of Cajal, also known as pacemaker cells and smooth muscle cells, coupled with gastric motility and contractions [4], maintaining a basal rhythm of 2–4 cpm. When there are episodes of gastric slow waves with different frequencies, they are termed bradygastria (0.5–2 cpm) and tachygastria (4–9 cpm) [10]. Moreover, the muscular wall is innervated by sympathetic splanchnic nerves and parasympathetic fibers controlling the electrical impulses. After food or beverage ingestion, the stomach fundus relaxes due to vagal efferent effect and acetylcholine release, which can cause coupling of gastric slow waves and spike impulses to produce gastric contractions [4,11]. The strong influence of parasympathetic and sympathetic activity on GMA causes a stress response characterized by elevated sympathetic activity and decreased parasympathetic activity. This leads to an interruption in gastric slow waves and increased tachygastria, shown in experimental studies with healthy subjects [4,12,13]. The prospects of using EGG to assess the normal and abnormal GMA and implications of gastric motility dysfunction have been gaining more interest.

A study conducted in 1991 reported that gastric spectral power is significantly lower in the obese than in those with normal weight among young and aged subjects. The study suggested that EGG spectral power could be influenced by the tissue thickness between the cutaneous electrode and the gastric muscular wall and, thus, by the adiposity of the participant [14]. However, Riezzo et al. reported that obesity did not affect EGG parameters of mean dominant frequency or the power ratio [15].

The meal composition has also been suggested to play an essential role because no difference in the gastric emptying time (normal at up to 4 h [16]) has been observed in studies conducted with solid meals containing protein or carbohydrate as the major caloric sources [14,17]. However, gastric emptying time is increased in obesity [16]. Unchanged gastric slow waves at baseline have been reported in obese subjects, but the responses have been enhanced by ingesting both protein and fats. The healthy-weight subjects demonstrated a lower percentage of normal gastric slow waves in response to a fatty meal, not observed in obese subjects. Moreover, following a protein meal, healthy-weight subjects showed no changes in the dominant frequency or power of the gastric slow waves, while both of these parameters were increased in obese subjects [18]. Autonomic functions were changed in obese subjects in fed and fasting conditions. Obese subjects also show higher sympathetic activity in the fasting state but the absence of a normal postprandial response in sympathovagal balance to both protein and fat [18].

In contrast, some studies reported that age, gender, and BMI do not affect GMA [8]. Still, few studies with adults, adolescents, and children have evaluated the relationship between dietary intakes and GMA in obesity to reach a decisive conclusion. Since GMA has been associated with the ANS and the ingested volume and composition of food, the use of diets based on certain macronutrient deficits, such as low-fat or low-carbohydrate diet, might be related to GMA changes. The aim of this study was to investigate the association of the GMA with reported dietary intakes, measured resting energy expenditure, and substrate utilization in participants with obesity.

## 2. Materials and Methods

### 2.1. Study Design and Sample Size

This study was a cross-sectional study with further analyses as case-control. It was conducted in the clinical nutrition clinic at the College of Applied Medical Science, King Saud University, Riyadh, SA. A sample size of 130 participants was calculated using the G* power software to meet 80% power, *p* < 0.05. Recruitment was conducted via an open invitation for participation which was announced via social media. The participants included in the study were aged 18 to 60 years, of both genders with no associated comorbidities, and not taking any medication affecting gastrointestinal function (such as prokinetic, anti-emetic, and non-steroidal anti-inflammatory medications), hydration status, or metabolic rate. In addition, they had no history of existing chronic diseases, such as cancer and heart diseases. All participants that were pregnant or had any psychiatric disorders and malignancies were excluded. The final number of participants was 115. The participants were divided into two groups based on their BMI: non-obese and obese. The two groups were further subdivided into four groups based on the body mass index: healthy weight (HW), overweight (OW), obesity (OB), and morbid obesity (MO) groups (Figure 1). 

The participants were asked to fast for 10–12 h before the assessment and avoid any medications that might affect gastric motility two days before the appointment. They were asked to refrain from drinking water for four hours before the appointment. The study was approved by the Research Ethics Review Board at King Saud University, Riyadh, Saudi Arabia, under reference number 20/0908/IRB. Before enrollment, participants were informed about the study’s purpose, methods, setting, and benefits and signed a consent form before participation.

### 2.2. Demographic Data

A self-reporting questionnaire was distributed among the participants to collect sociodemographic data and medical history. The socio-demographic data comprised gender, marital status, education status, physical activity, and smoking.

### 2.3. Anthropometric Measurement

A scale was used to manually measure weight and height (Seca Co., Hamburg, Germany). The weight was recorded in kilograms (Kg) and the height was measured approximately to the nearest one centimeter. BMI was calculated as weight divided by the square of height (kg/m^2^). The participants were divided into groups based on BMI: healthy weight (18.5–24.9 kg/m^2^), overweight (25–29.9 kg/m^2^), obese (30–39.9 kg/m^2^), and morbidly obese (≥40 kg/m^2^).

### 2.4. Resting Energy Expenditure

The measured RMR (RMR_m) was examined by automated metabolic cart indirect calorimetry (IC) using a standard canopy hood (Quark RMR, Cosmed, Italy). This device measured the rates of O_2_ consumption (VO_2_) and CO_2_ production (VCO_2_), consequently calculating the respiratory quotient (RQ), percentages of each substrate utilization, and the resting energy expenditure (REE) using the Weir equation (REE = (3.94 × VO_2_) + (1.1 × VCO_2_) [19]. The Quark device was initially warmed up for 20 min after being powered on and calibrated. The turbine and gas calibration was conducted every day before the participants attended the clinic. The antibacterial filter changed daily and a single-use disposable plastic cover was used for the canopy. The patient was instructed to rest in a 30° head-up position for 15 min without movement or sleep during the test. The data collected from indirect calorimetry were RMR_m (Kcal/day), predicted RMR by using the Harris–Benedict equation (RMR_pred; Kcal/day), percent of prediction in the equation, variability percentage, oxygen consumption VO_2_ mL/m, carbon dioxide production VCO_2_ mL/m, respiratory quotient (RQ) = VCO_2_/VO_2,_ percentage of fat, carbohydrates, and protein utilization. All measurements were performed in the morning period between 8:00 and 11:30 a.m.

### 2.5. Dietary Assessment

Dietary assessment was reported using the multiple-pass 24 h recall for two days preceding the test day. The assessment was completed on the examination day by licensed clinical dietitians. The food models were used to estimate dietary intake and lower the misestimated portion size. The dietary data were analyzed using the ESHA’s Food Processor® software version 11.11.32 (Esha Research Inc, Salem, OR, USA). This software has an extensive food and nutrition database, which allows for analyzing and comparing the macro- and micronutrients between study groups. It also allows for inserting new recipes, based on the ingredients, providing analysis for the entered food. The data were then reported as a spreadsheet containing details of macro- and micronutrients consumed with the average intake and then compared to the recommendation.

### 2.6. EGG for Measuring the GMA Activity

The GMA was measured by a multichannel electrogastrography (EGG) device with a satiety water-load test (3CPM Company, Sparks, MD, USA). The EGG has three electrodes placed on the epigastrium skin and a respiratory belt placed across the upper chest and under the armpits. The placement of the electrodes is detailed as follows. The red EGG electrode was positioned in the mid-clavicular line, approximately two inches below the left costochondral margin. The green EGG electrode was positioned in the mid-clavicular line on the right side, two inches below the right costochondral margin, and the black EGG electrode was positioned between the xiphoid and umbilicus in the midpoint.

The participants were positioned in a comfortable supine position, with dim lighting, and allowed no movement or talking to avoid motion artifacts. The skin was shaved clear of hair and made dry before placing the EGG electrodes and respiratory belt and the test was recorded for 5 min. Once the EGG and respiratory signals were stable, the baseline (pre-prandial) EGG recorded period began. The record duration was approximately 40 min, including Baseline (pre-prandial) EGG recording for 10 minutes. This was followed by the participants sitting up for drinking a maximally tolerated water load within 5 min and the volume of ingested water was recorded in the EGG software. The participant then resumed the supine position and the post-prandial EGG was recorded for 30 min. The respiratory rate was assessed and recorded every 10 min by counting the chest raises for one minute. The EGG and respiratory records were reported as four periods, baseline, and every 10 min was presented as one period (i.e., BL, Min10, Min20, and Min30, respectively). The parameters obtained from the EGG included the volume of water load ingested. Distribution of average power by frequency region as a percentage of power in the 0 to 15 CPM range; the bradygasteria (1.0–2.5 CPM), normogastria (2.5–3.75 CPM), tachygasteria (3.75–10 CPM), and duodenal (10–15 CPM) during the four periods. Moreover, the average dominant frequency (ADF) was used to analyze the association between the reported dietary intake and GMA.

### 2.7. Statistical Analysis

The statistical analysis was performed using the Statistical Package for the Social Sciences (SPSS, version 25, Chicago, IL, USA). The study parameters are presented as means ± SD, and the Shapiro–Wilk test was used to test the normality of the parameters. The ANOVA with a post hoc test was used to compare the means of the four subgroups (HW, OW, OB, and MO groups). The Pearson correlation coefficient assessed the association of the ADF with dietary intakes and parameters of indirect calorimetry. The correlations were tested among the four subgroups and between participants with and without obesity. All differences were considered significant if the *p*-value ≤ 0.05.

## 3. Results

### 3.1. Demographic and Descriptive Results

Table 1 presents the demographic characteristics of participants stratified by subgroups. There were no differences in marital status, education level, smoking, and physical activity between groups; however, male participants represented 55.7% of the total sample, while 44.3% were female (*p* = 0.01) (Table 1).

### 3.2. Differences in Indirect Calorimetry and Other Data among Subgroups

Mean (±SD) differences in indirect calorimetry data for each subgroup are presented in Table 2. There was a significant difference in age among subgroups; younger participants were mostly in the non-obese group, while older participants were in the obese group. There was a significant difference in the measured RMR_m, RMR_pred, and VO_2_ among subgroups (*p* < 0.001). In addition, VCO_2_ was significantly higher in the obese than in the non-obese group (*p* < 0.001). The MO group significantly utilized more fat than the HW and OW (*p* < 0.05); this difference was not observed in the OB group. On the other hand, the percent of protein utilization was significantly lower in the OB and MO groups compared to the HW and OW groups (*p* < 0.001). There was no difference in carbohydrate utilization between subgroups (Table 2).

### 3.3. Differences in Dietary Intake among Subgroups

Appendix A represents the differences in macro/micronutrient intakes and trace minerals among subgroups. Macronutrient intakes (protein, carbohydrate, and fat) were higher in the MO group compared to HW (*p* < 0.01). No significant differences between groups were observed for monosaccharide, disaccharide, PUFA, trans fat, and cholesterol. Only iron, K, and Na differed significantly between groups. No differences were noted for other minerals/vitamins (Appendix A).

### 3.4. EGG Recordings among Subgroups

The EGG recording at Min20 was significant in normogastria (normoG) percentages, lower in the OB group compared to other groups (Table 3). In addition, the percentage of duodenal rhythm at Min20 was higher in the OB group than in other groups. Compared to the OB group, the percentage of duodenal rhythm at Min30 was significantly higher in the MO group (10.93 ± 14.65 vs. 3.72 ± 2.78, *p* < 0.05) (Table 3).

### 3.5. Correlations of the Average Dominant Frequency of the EGG with Dietary Intakes and Energy Expenditure

Total daily kcal intake, fat, and carbohydrate intakes were positively correlated with ADF at the first 10 min in the OW group (r = 0.49, 0.43, 0.42, respectively, *p* < 0.05) (Table 4). There was no correlation between dietary intake and ADF in the first 10 min in the HW group.

Protein intake was directly correlated with ADF at 20–30 min (r = 0.56, *p* < 0.05) in the HW group, but not in the other groups. In addition, protein intake was correlated with water-load volume in the HW group (r = 0.41, *p* < 0.05), while CHO intake was positively correlated with water-load volume in the MO group (r = 0.49, *p* < 0.05) but not in the OB group (Table 4).

Regarding correlations of indirect calorimetry parameters and EG, there were no significant correlations in the HW group (Table 5). However, in the OW group, RQ was positively correlated with ADF in the first 30–40 min (r = 0.68, *p* < 0.05), percent fat showed a positive correlation with water-load volume and a negative correlation with ADF in the first 30–40 min (r = 0.37, −0.68, respectively, *p* < 0.05), percent CHO utilization showed a positive correlation with ADF in the first 30–40 min (r = 0.66, *p* < 0.05), and percent protein showed a negative correlation with water-load volume (r = −0.40, *p* < 0.05) (Table 5). In the OB group, the RMR_m correlated positively with the volume of the stomach determined by water load (r = 0.37, *p* < 0.05). In addition, ADF at the first 10–20 min showed a positive correlation with the percent of fat utilization and a negative correlation with both the RQ and the percent of CHO utilization (Table 5). Moreover, percent of protein utilization showed negative correlations with ADF in the first 20–30 and 30–40 min (r = −0.44, −0,46, respectively, *p* < 0.01). In the MO group, correlations were non-significant (Table 5).

### 3.6. Correlations in Study Participants Assigned into Two Groups Only (Non-Obese vs. Obese Groups)

In order to avoid the negative effect of the subdivision on the significant correlation, Table 6 shows the correlations in the non-obese and obese groups. In the non-obese group, total calorie, fat, carbohydrate, and total fiber intakes showed significant correlations with the ADF in the first 10 min, while protein intake showed a significant correlation with ADF at 20 to 30 min (Table 6). This was not the case in the obese group, i.e., total fiber intake was the only factor having significant positive correlations with ADF at the three postprandial recording periods (r = 0.33, 0.38, and 0.29, respectively, *p* < 0.05). In addition, CHO intakes showed a positive correlation with the gastric volume determined by water loading.

Regarding the indirect calorimetry parameters, the percentage of fat utilization in the non-obese group showed an inverse correlation with the ADF at 30 to 40 min, while the percentage of CHO utilization had a positive correlation with the ADF at the same period of recording. In those with obesity, measured RMR showed a positive correlation with the gastric volume determined by water loading (r = 0.39, *p* < 0.05). Moreover, the percentage of protein utilization has an inverse correlation with the ADF at 20 to 30 and 30 to 40 min (Table 6).

## 4. Discussion

The present study is one of the few to investigate the associations between the GMA—represented by an EGG—and the dietary intake, substrate utilization, and energy expenditure of healthy-weight, overweight, obese, and morbidly obese participants. We found that total daily calorie (kcal), fat, carbohydrate, and fiber intakes positively correlated with ADF at early postprandial (10 min) in the no-obesity groups and the OW subgroup. Carbohydrates were also positively correlated with water load in the obesity groups and the MO subgroup. We observed a positive correlation between water load and ADF at 20–30 min for protein intake in the HW subgroup. In the obesity group and the OB subgroup, RMR_m was positively correlated with water-load volume. ADF during 20–30 min showed a positive correlation with the percent of fat utilization and a negative correlation with the RQ and percent of CHO utilization in the OB subgroup.

Food intake has been documented to alter the characteristics of GMA [20]. However, limited data are available on the influence of the types of macronutrients on GMA. The influence of water-load testing on individuals with obesity has also been minimally investigated. Only a few studies have been conducted on the fed state, showing no significant difference in the dominant power after meals among healthy-weight adults [21] and obese children [22]. Furthermore, data on whether obesity impairs gastrointestinal motility are inconsistent, with some reports showing faster, unchanged, or delayed gastric emptying [18,23,24]. In our study, we reported that fat intake was positively correlated with ADF in early postprandial (10 min) in the OW subgroup, suggesting that individuals with higher weight adapted better to fatty meals than their counterparts. Furthermore, in contrast to the positive correlation between protein and ADF at 20–30 min in the HW subgroup found in our results, the previously mentioned trial [18] showed no postprandial alterations in the percentage of normal slow waves in both lean and obese subjects after a protein-rich meal. They also reported a decreased dominant power of the EGG in obese individuals compared to lean individuals and a postprandial increase in slow-wave frequency. This suggests gastric motility is more responsive to protein intake in obese individuals, which explains why they can tolerate fat- and protein-rich meals compared to lean individuals [18]. Moreover, an earlier study compared the GMA across different ages, genders, and BMI levels [25]. It reported that in overweight (BMI > 25 kg/m^2^) individuals, there was an overall decrease in the absolute dominant frequency power but a similar increase in the postprandial dominant frequency power. Overweight individuals had a postprandial decrease in the slow-wave coupling compared to those with a healthy weight (BMI < 25 kg/m^2^), suggesting that a higher BMI reduces the dominant postprandial power [25]. In the present study, we also found that carbohydrates positively correlated with ADF during the first 10 min in the no-obesity and OW subgroups. A few studies compared the responses of obese and non-obese individuals to a carbohydrate-rich meal. Following the consumption of test meals with different glycemic indexes in rats, a significant inverse correlation was seen between the normality of intestinal slow waves and blood glucose [26]. In humans, high-carbohydrate, low-fat isocaloric meals were given to healthy individuals. Meals high in carbohydrates and low in fat significantly increased the EGG [27]. We also reported that carbohydrates were positively correlated with water load in the obesity groups. The fiber content in the diet is also important. In the current study, total fiber intake was positively associated with the ADF in participants with obesity, which provides a basis for the beneficial effect of fiber-containing diets [28]. Additionally, the adiposity level may affect the EGG spectral ability, as a thicker abdominal wall increases the distance between the cutaneous electrode and the recording electrodes [29]. The varying results from a few studies may be attributed to several factors, e.g., the different test meals applied—including samples of varying age and gender—and the duration and level of obesity. Furthermore, studies applied different methods for assessing GMA. Meal-induced alterations in the autonomic function may also have contributed to the variance in findings.

Interestingly, concerning energy expenditure, RQ was inversely correlated with ADF in the first 10–20 min in the OB subgroup. This finding is interesting as this correlation is concurrent with the positive correlation between the ADF in the early postprandial stage and the fat and the negative correlation with carbohydrates we observed in the OB individuals. Whether GMA impacts energy expenditure is unclear, as no studies were found in this regard, and further investigations are needed to confirm these findings.

Our study is one of the few to investigate the association of different nutrients with GMA in participants with different weight statuses. We included participants of four different weight statuses and measured the average dominant frequency of the EGG. We took extensive measures to increase data reporting accuracy; for example, we incorporated detailed two-day, multiple-pass 24 h recalls by trained dietitians. Furthermore, the RMR_m was measured using the automated metabolic cart (indirect calorimetry) with a standard canopy hood. However, the study is not without its share of limitations. The observational design of the current study does not allow causality to be determined. Moreover, we did not use test meals of different macronutrients during the study. Therefore, comparing our findings with current RCTs would be difficult, as we did not measure gastric motility after test meals. Despite our intention to reduce error, recall bias is unavoidable. Furthermore, although some participants tend to talk, move their hands, or adjust their body posture during the EGG test, these motion artifacts are unlikely to affect our findings because the computerized spectral analysis is adjusted for those artifacts.

## 5. Conclusions

In conclusion, the GMA showed distinctive patterns in the obese and non-obese groups, which may be influenced by different degrees of the characteristics and compositions of meals among obese and non-obese individuals. These findings support the approach that influencing ADF of the stomach via certain food items may be used as a basis for tailoring some dietary regimens for obesity management, given that gastric motility is a key mediator of hunger, satiation, and satiety and may also play a role in the long-term regulation of body weight. Moreover, GMA is linked with substrate utilization and energy expenditure. For example, dietary regimens with high fiber intake may improve the ADF of the stomach in cases of obesity, which is characterized by bradygasteria. Further research should be conducted to understand the underlying mechanisms and to confirm these findings.

## Figures and Tables

**Figure 1 nutrients-14-04021-f001:**
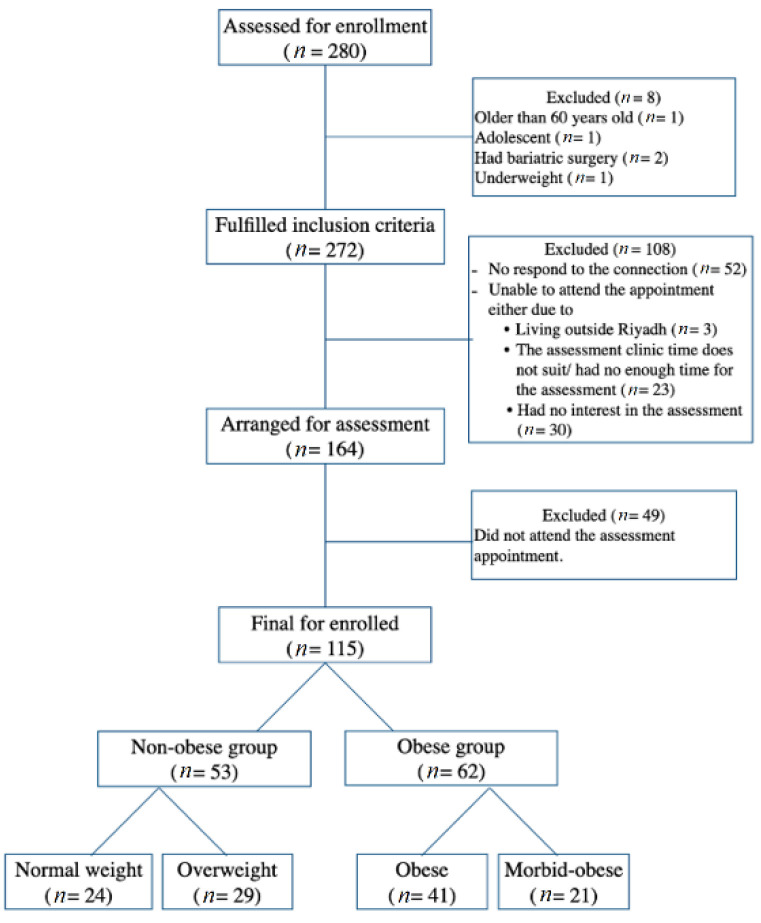
A systematic representation of the participants’ allotments to different groups based on BMI.

**Table 1 nutrients-14-04021-t001:** Demographic characteristics of participants stratified by subgroups, *n* = 115.

Variables	Non-Obese				Obese				*p*-Value
	HW (*n* = 24)		OW (*n* = 29)		OB (*n* = 41)		MO (*n* = 21)		
	%within Gender	%within Subgroup	%within Gender	%within Subgroup	%within Gender	%within Subgroup	%within Gender	%within Subgroup	
Gender									0.01 *
Female	33.33	70.80	23.53	41.38	33.33	41.46	9.80	23.81	
Male	10.94	29.17	26.56	58.62	37.50	58.54	25	76.19	
Smoking									0.53
Not smoker	19.28	84.21	27.71	82.14	36.15	93.75	16.87	82.35	
Smoker	23.08	15.79	38.46	17.85	15.39	6.25	23.08	17.65	
Physical activity									0.11
Active	13.85	40.91	29.23	65.52	32.31	56.76	24.62	76.19	
Inactive	29.55	59.09	22.73	34.48	36.36	43.24	11.36	23.81	

HW, healthy weight; OW, overweight; OB, obesity; MO, morbid obesity. * *p* < 0.05 (2-tailed).

**Table 2 nutrients-14-04021-t002:** Mean differences in indirect calorimetry and other descriptive data stratified by subgroups, *n* = 115.

Variables	HW (*n* = 24) Mean ± SD	OW (*n* = 29) Mean ± SD	OB (*n* = 41) Mean ± SD	MO (*n* = 21) Mean ± SD	*p*-Value
Age	24.46 ± 7.44 ^a^	29.07 ± 9.80 ^a,b^	30.98 ± 10.74 ^b,c^	34.90 ± 11.32 ^c^	0.006 *
BMI	21.86 ± 1.91 ^a^	27.69 ± 1.16 ^a^	35.32 ± 3.36 ^a^	44.08 ± 4.11 ^a^	<0.001 *
RMR_m (kcal/day)	1412.92 ± 202.64 ^a^	1578.17 ± 268.81 ^b^	1835.85 ± 326.08 ^c^	2222.38 ± 334.25 ^d^	<0.001 **
RMR_pred (kcal/day)	1497.96 ± 187.24 ^a^	1678.59 ± 238.01 ^b^	2004.63 ± 373.47 ^c^	2392.19 ± 338.37 ^d^	<0.001 **
Prediction %	94.52 ± 8.82 ^a^	94.56 ± 13.80 ^a^	92.50 ± 11.16 ^a^	93.71 ± 12.77 ^a^	0.87
Variability %	6.96 ± 1.79 ^a^	8.88 ± 2.44 ^b^	8.15 ± 2.06 ^b,c^	7.71 ± 1.35 ^a,c^	0.007 *
RQ	0.84 ± 0.07 ^a^	0.83 ± 0.07 ^a b^	0.83 ± 0.05 ^a,b^	0.81 ± 0.05 ^b^	0.28
VO_2_	204.75 ± 29.61 ^a^	228.93 ± 38.62 ^b^	266.17 ± 47.07 ^c^	323.71 ± 49.08 ^d^	<0.001 **
VCO_2_	172.58 ± 25.19 ^a^	190.66 ± 35.0 ^a^	220.54 ± 40.27 ^b^	260.57 ± 38.66 ^c^	<0.001 **
Fat %	40.05 ± 18.12 ^a^	44.90 ± 21.27 ^a^	47.24 ± 18.31 ^a b^	56.63 ± 16.73 ^b^	0.03 *
CHO %	36.02 ± 17.96 ^a^	32.76 ± 21.04 ^a^	34.48 ± 17.90 ^a^	28.10 ± 16.08 ^a^	0.50
Protein %	23.91 ± 3.30 ^a^	22.34 ± 4.34 ^a^	18.29 ± 4.53 ^b^	15.26 ± 2.29 ^b^	<0.001 **

Different superscripts (a, b, c, and d) indicate statistically different. HW, healthy weight; OW, overweight; OB, obesity; MO, morbid obesity, BMI, body mass index; RMR_m: measured resting metabolic rate by using indirect calorimetry; RMR Pred.: resting metabolic rate predicted; RQ: respiratory quotient; VO_2_: oxygen consumption; VCO_2_: carbon dioxide production. * *p* < 0.05 (2-tailed); ** *p* < 0.001 (2-tailed).

**Table 3 nutrients-14-04021-t003:** EGG patterns among study subgroups stratified by subgroups, *n* = 115.

Variables	HW (*n* = 24) Mean ± SD	OW (*n* = 29) Mean ± SD	OB (*n* = 41) Mean ± SD	MO (*n* = 21) Mean ± SD	*p*-Value
Water-load	509.17 ± 182.40 ^a^	514.48 ± 216.94 ^a^	512.88 ± 244.35 ^a^	774.29 ± 388.07 ^b^	0.001 **
BL-BradayG	48.10 ± 18.93 ^a^	50.22 ± 20.35 ^a^	51.05 ± 25.10 ^a^	66.86 ± 15.08 ^b^	0.014 *
BL-NromoG	24.86 ± 18.85 ^a^	13.51 ± 8.62 ^b^	11.23 ± 7.42 ^b^	13.49 ± 7.28 ^b^	<0.001 **
BL-TachyG	18.39 ± 7.81 ^a^	21.02 ± 13.15 ^a^	21.57 ± 16.69 ^a^	14.83 ± 9.25 ^a^	0.242
BL-Duodenal	8.62 ± 6.96 ^a,c^	15.25 ± 16.55 ^a,b^	16.15 ± 16.90 ^b^	5.30 ± 4.84 ^c^	0.01 *
Min10_BradyG	58.48 ± 18.34 ^a,b^	57.97 ± 15.22 ^a,b^	49.80 ± 22.46 ^a^	62.08 ± 16.28 ^b^	0.069
Min10_NormoG	14.20 ± 8.35 ^a^	13.88 ± 6.74 ^a^	11.29 ± 6.62 ^a^	15.27 ± 11.97 ^a^	0.257
Min10_TachyG	19.83 ± 13.17 ^a^	20.37 ± 9.59 ^a^	25.29 ± 13.96 ^a^	17.88 ± 11.87 ^b^	0.106
Min10_Duodenal	7.48 ± 7.86 ^a^	7.78 ± 7.49 ^a^	13.61 ± 12.84 ^b^	4.77 ± 4.78 ^a^	0.003 *
Min20_BradyG	48.98 ± 22.73 ^a^	50.92 ± 14.81 ^a^	46.65 ± 19.19 ^a^	58.33 ± 16.46 ^b^	0.135
Min20_NormoG	23.85 ± 18.34 ^a^	20.37 ± 12.64 ^a^	13.96 ± 6.39 ^b^	18.49 ± 15.26 ^a,b^	0.024 *
Min20_TachyG	19.83 ± 10.92 ^a^	21.24 ± 9.30 ^a,b^	25.89 ± 13.73 ^b^	18.06 ± 8.23 ^a^	0.042
Min20_Duodenal	7.35 ± 8.79 ^a^	7.47 ± 8.95 ^a^	13.50 ± 16.50 ^b^	5.11 ± 5.15 ^a^	0.031 *
Min30_BradyG	48.90 ± 20.50 ^a,b,c^	51.18 ± 16.17 ^a,b,c^	46.63 ± 19.24 ^b^	58.39 ± 16.19 ^c^	0.119
Min30_NormoG	21.91 ± 15.90 ^a^	19.53 ± 12.20 ^a^	20.16 ± 14.51 ^a^	21.13 ± 16.06 ^a^	0.937
Min30_TachyG	21.40 ± 11.25 ^a^	22.54 ± 9.51 ^a^	22.28 ± 12.78 ^a^	16.75 ± 8.33 ^a^	0.238
Min30_Duodenal	7.79 ± 7.90 ^a,b,c^	6.75 ± 5.54 ^a,b,c^	10.93 ± 14.65 ^b^	3.72 ± 2.78 ^c^	0.054 *
ADF-10 min	1.74 ± 0.81 ^a,b^	2.29 ± 1.73 ^a,b^	2.70 ± 3.11 ^a^	1.29 ± 0.44 ^b^	0.069
ADF-20 min	1.90 ± 1.19 ^a,b^	1.62 ± 0.52 ^a^	2.80 ± 3.33 ^b^	1.49 ± 0.45 ^a^	0.047 *
ADF-30 min	2.53 ± 2.23 ^a^	1.74 ± 1.02 ^a^	2.56 ± 2.62 ^a^	1.60 ± 0.70 ^a^	0.141
ADF-40 min	1.61 ± 0.96 ^a,b^	1.6 ± 0.76 ^a,b^	2.36 ± 3.02 ^a^	1.16 ± 0.68 ^b^	0.105

Different superscripts (a, b and c) indicate statistically different. BadyG, bradygasria; BL, baseline; NormoG, normogastria; TachyG, tachygastria. ADF, Average dominant frequency. * *p* < 0.05 (2-tailed); ** *p* < 0.001 (2-tailed)

**Table 4 nutrients-14-04021-t004:** Pearson correlation coefficient between dietary intake and gastric myoelectrical activity among subgroups, *n* = 115.

	HW (*n* = 24)				OW (*n* = 29)				OB (*n* = 41)				MO (*n* = 21)			
	Kcal	Fat	CHO	Pro.	Kcal	Fat	CHO	Pro.	Kcal	Fat	CHO	Pro.	Kcal	Fat	CHO	Pro.
Water load	0.18	−0.28	0.29	0.41 *	−0.04	0.05	−0.25	0.04	−0.07	−0.11	−0.03	−0.07	0.35	0.17	0.49 *	0.14
The ADF at first 10 min.	−0.21	−0.01	−0.08	−0.21	0.49 *	0.43 *	0.42 *	0.36	−0.19	0.01	−0.17	−0.26	−0.13	−0.27	−0.09	−0.06
The ADF at 10–20 min.	0.16	0.12	−0.33	−0.18	−0.05	−0.13	0.13	−0.10	0.12	0.19	0.25	−0.19	−0.18	−0.19	−0.20	−0.01
The ADF at 20–30 min.	0.15	−0.34	0.15	0.56 *	−0.10	−0.25	0.10	−0.09	0.09	0.10	0.22	−0.15	0.19	0.10	0.29	0.01
The ADF at 30–40 min.	0.16	−0.13	0.34	0.18	−0.28	−0.29	−0.12	−0.26	0.11	0.12	0.19	−0.15	0.24	0.31	0.26	0.29

ADF, average dominant frequency; CHO, carbohydrate; Kcal, calorie; MO, morbidly obese; HW, Healthy weight; OW, overweight; OB, obese; Pro, protein. * *p* ≤ 0.05.

**Table 5 nutrients-14-04021-t005:** Correlation between parameters of energy expenditure and gastric myoelectrical activity among subgroups, *n* = 115.

	HW (*n* = 24)					OW (*n* = 29)					OB (*n* = 41)					MO (*n* = 21)				
	RMR_m	RQ	Fat %	CHO %	Prot %	RMR_m	RQ	Fat %	CHO%	Prot %	RMR_m	RQ	Fat %	CHO%	Prot %	RMR_m	RQ	Fat %	CHO%	Prot %
Water load Volume	−0.04	0.14	−0.17	0.17	0.01	0.32	−0.32	0.37 *	−0.29	−0.40 *	0.37 *	0.14	−0.12	0.16	−0.16	0.13	−0.22	0.25	−0.24	−0.16
The ADF at first 10 min.	−0.18	−0.23	0.20	−0.23	0.16	013	−0.18	0.25	−0.22	−0.17	−0.06	−0.11	0.11	−0.09	−0.08	0.06	−0.06	0.02	−0.01	−0.07
The ADF at 10–20 min.	0.10	0.31	−0.27	0.29	−0.13	−0.13	0.27	−0.33	0.28	0.26	0.02	−0.36 *	0.37 *	−0.32 *	−0.23	−0.34	0.02	−0.06	0.02	0.33
The ADF at 20–30 min.	−0.27	0.19	−0.23	0.19	0.26	−0.10	0.15	−0.12	0.11	0.08	0.12	−0.29	0.06	−0.24	−0.44 **	0.05	−0.10	0.11	−0.10	−0.06
The ADF at 30–40 min.	0.01	−0.04	0.03	−0.04	0.04	0.01	0.68 *	−0.68 *	0.66 *	0.14	0.08	0.01	0.15	0.05	−0.46 **	−0.13	0.17	−0.18	0.17	0.17

ADF, average dominant frequency; RMR, resting metabolic rate; RQ, respiratory quotient; Fat%, percent of fat utilization during measurement; CHO%, percent of carbohydrate measurement during measurement; Prot %, percent of protein utilization during measurement; HW, Healthy weight; OW, overweight; OB, obese; MO, morbidly obese.* *p* ≤ 0.05; ** *p* ≤ 0.01.

**Table 6 nutrients-14-04021-t006:** Pearson Correlation coefficient between gastric myoelectrical activity and dietary intakes as well as parameters of indirect calorimetry in study participants assigned as two groups (Non-obese and Obese groups).

	Non-Obese (*n* = 53)					Obese (*n* = 62)					Non-Obese (*n* = 53)					Obese (*n* = 62)				
	Kcal	Fat	CHO	Pro.	Fiber	Kcal	Fat	CHO	Pro.	Fiber	RMR_m	RQ	Fat %	CHO %	Prot %	RMR_m	RQ	Fat %	CHO %	Prot %
Water load	0.04	−0.07	−0.04	0.20	0.08	0.18	0.02	0.28 *	0.01	0.10	0.19	−0.14	0.17	−0.12	−0.26	0.39 *	−0.08	0.13	−0.07	−0.26
The ADF at first 10 min.	0.32 *	0.30 *	0.31 *	0.16	0.29 *	−0.15	−0.01	−0.13	−0.18	0.06	0.12	−0.20	0.25	−0.23	−0.13	−0.17	−0.04	0.02	−0.03	0.02
The ADF at 10–20 min.	0.06	0.03	0.19	−0.14	0.03	0.05	0.13	0.13	−0.13	0.33 *	−0.05	0.28 *	−0.27 *	0.26	0.06	−0.12	−0.24	0.23	−0.21	−0.11
The ADF at 20–30 min.	0.03	−0.24	0.07	0.33 *	0.05	0.07	0.09	0.16	−0.10	0.38 *	−0.26	0.18	−0.19	0.16	0.20	−0.02	−0.20	0.23	−0.16	−0.31 *
The ADF at 30–40 min.	−0.08	−0.21	0.10	−0.02	0.06	0.08	0.12	0.13	−0.07	0.29 *	−0.01	0.32 *	−0.34 *	0.33 *	0.09	−0.07	0.05	−0.02	0.10	−0.30 *

ADF, average dominant frequency; CHO, carbohydrate; Kcal, calorie; Pro, protein. ** p* ≤ 0.05.

## Data Availability

Original data supporting these results are available on request from the corresponding author for reasonable purposes.

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
