# Peer review of "Association of Gastric Myoelectric Activity with Dietary Intakes, Substrate Utilization, and Energy Expenditure in Adults with Obesity"

_nutrients, 2022, doi:10.3390/nu14194021_

Round 1

Reviewer 1 Report

Interesting exploration of the difference in gastric motility amongst individuals of different weight status, and how this is associated with energy expenditure and dietary intake.
Abstract
- ADF - Write in full at first mention
Intro
- Line 74 – 75 – Is this suggesting that EGG is not an appropriate assessment method of GMA
for participants who are obese? As the tissue thickness influences measurements and therefore does not accurately reflect the GMA. Please clarify.
- Please highlight the clinical significance of gastric emptying time/ GMA / gastric slow waves. Is it beneficial for this to be high or now?
Methods
- If possible, please change “normal weight” to “healthy weight” throughout manuscript
- Were participants who previously had gastric surgery excluded?
- Line 118 – 119 – is this ethical approval? Please state explicitly
- Line 130 – write cm in full
- What is the difference between RMR and pre-RMR?
- What time of day were the visits conducted – this has an independent effect on RMR
- It may be useful for the reader to know the sequence of measurements conducted on the visit day – e.g. was RMR measured first or GMA?
Results
- Marital status and education level is not associated with gastric emptying rate, there’s no need to report on this in Table 1.
- Line 215 – possible spelling error – Pred.RER
- Table 2 – protein was not abbreviated, no need to appear in footnote
- Line 218 – 220 - % of protein utilization does not seem higher in the obese group compared to the non-obese group, according to Table 2.
- Table 3 – data for micronutrients can go into a supplementary table, unless there is evidence to suggest that gastric motility will affect micronutrient intake.
- What does min-20 stand for? Is this 20 minutes before baseline? Or 20 minutes postprandial?
Discussion
- Line 296 - Should read normal “weight” adults
- Line 308 – 310 – attempt to explain what your results mean, rather than the results of reference 14
- Reference 4 is not appropriate for comparison as there were no children included in this study
- Line 334 – 343 – there is minimal clinical significance to the findings reported in this paragraph
- A focus on clinical significance is required in the discussion. If people with different weight have different gastric motility rates and substrate utilization, how does this information influence practice?

Author Response

Response to Reviewer 1

Manuscript ID:  nutrients-1832948

Manuscript Title: ‘Association of gastric myoelectric activity with dietary intakes, substrates utilization, and energy expenditure in adults with obesity

We thank the reviewers for their careful examination of the manuscript and appreciate the useful suggestions to improve the quality of our paper. Our point-by-point response to the reviewers' comments is given below. Changes in the manuscript are indicated in red font. Please note that the pages and line numbers mentioned in the reviewers’ comments refer to the original manuscript, whereas those in the authors’ reply refer to the revised manuscript.

Comments from the Editors and Reviewers:

  1. ADF - Write in full at first mention

Thank you for pointing this out. The ADF full name has been added (line 30)

  1. Line 74 – _75 – _Is this suggesting that EGG is not an appropriate assessment method of GMA for participants who are obese? As the tissue thickness influences measurements and therefore does not accurately reflect the GMA. Please clarify.

Answer: This part has been clarified and supported with a reference (lines 85-86)

  1. Please highlight the clinical significance of gastric emptying time/ GMA / gastric slow waves. Is it beneficial for this to be high or low?

Answer: This part has been clarified and supported with references (line 88, 90), (lines 63-64), (lines 69-71)

  1. If possible, please change “normal weight” to “healthy weight” throughout the manuscript

Answer: Changes have been made throughout the text and tables (lines: 25-26, 29, 92, 94, 121, 143, 327)

  1. Were participants who previously had gastric surgery excluded?

Answer: Gastric surgeries especially bariatric surgeries that involve the stomach produce major anatomical changes in the stomach with disturbance of the GMA due to excision of interstitial cells of Cajal with variable degrees. For avoiding the bias of surgery we excluded patients with a history of gastric surgeries.

  1. Line 118 – _119 – _is this ethical approval? Please state explicitly

Answer: This is now clarified in line 130

  1. Line 130 – _write cm in full

Answer: This is now clarified in line 141

  1. What is the difference between RMR and pre-RMR?

Answer: RMR is the measured RMR (RMR_m) which was done by the automated metabolic cart indirect calorimetry, while pred RMR is the predicted RMR by using the Harris-Benedict equation (RMR_pred). both were clarified in section 2.4 and changed throughout the manuscript.

  1. What time of day were the visits conducted – _this has an independent effect on RMR

 Answer: thanks for this important question. All measurements were done in the morning period between 8:00 and 11:30 am. This was added to the text in lines 160-161.

  1. It may be useful for the reader to know the sequence of measurements conducted on the visit day – _e.g. was RMR measured first or GMA?

Answer: the procedures were done in the same order in the methods section of the manuscript. i.e. indirect calorimetry was done first then the EGG. The EGG testing was made at the end as it encounters drinking plenty amount of water and staying in the bed for about 45 minutes.

  1. Marital status and education level is not associated with gastric emptying rate, there’s no need to report on this in Table 1.

Answer: These are now excluded from Table 1

  1. Line 215 – _possible spelling error – _Pred.RER

Answer: This is now corrected in line 227

  1. Table 2 – _protein was not abbreviated, no need to appear in a footnote

Answer: This is now removed from the footnote

  1. Line 218 – _220 - % of protein utilization does not seem higher in the obese group compared to the non-obese group, according to Table 2.

Answer: This is now corrected in line 231

  1. Table 3 – _data for micronutrients can go into a supplementary table, unless there is evidence to suggest that gastric motility will affect micronutrient intake.

Answer: Table 3 is now table S1, all subsequent tables have been renumbered.

  1. What does min-20 stand for? Is this 20 minutes before baseline? Or 20 minutes postprandial?

Answer: It means the postprandial period from the 10th to 20th minutes of recording. It is now clarified in lines 193-194

  1. Line 296 – Should read normal ‘weight’ adults

Answer: This is now corrected in line 307

  1. Line 308 – _310 – _attempt to explain what your results mean, rather than the results of reference 14

Answer: This is now corrected in lines 312-314

  1. Reference 4 is not appropriate for comparison as there were no children included in this study

Answer: This reference has been removed from this paragraph

  1. Line 334 – _343 – _there is minimal clinical significance to the findings reported in this paragraph

Answer: This paragraph has been removed

  1. A focus on clinical significance is required in the discussion. If people with different weights have different gastric motility rates and substrate utilization, how does this information influence practice?

Answer: Clinical implications have been added to lines 365-369

Reviewer 2 Report

While the subject of this study is generally and clinically relevant, the authors tended to explain most results in a descriptive way. The causalities between GMA and dietary intakes, substrate utilization, and energy expenditure in obese subjects were not well defined and supported by a clear hypothesis.

In this study, the significant correlations between different parameters seemed to be observed randomly observed in a specific group. There was no consistent correlation of a particular parameter established with the progression of the severity of obesity, detract significantly from the conceptual novelty of the findings.

Author Response

Response to Reviewer 2

Manuscript ID:  nutrients-1832948

Manuscript Title: “Association of gastric myoelectric activity with dietary intakes, substrates utilization, and energy expenditure in adults with obesity”

We thank the reviewers for their careful examination of the manuscript and appreciate the useful suggestions to improve the quality of our paper. Our point-by-point response to the reviewers' comments is given below. Changes in the manuscript are indicated in red font. Please note that the pages and line numbers mentioned in the reviewers’ comments refer to the original manuscript, whereas those in the authors’ reply refer to the revised manuscript.

Comments from the Editors and Reviewers:

  1. Are the results clearly presented?

Must be improved.

Response:  Based on reviewer 1 and your suggestion we changed the order and clarity of results. For example Table 3 is now table S1.

  1. Are the conclusions supported by the results?

Must be improved.

Response: It is now improved.

  1. I While the subject of this study is generally and clinically relevant, the authors tended to explain most results in a descriptive way.

Response: As an association study with a case-control design, the results are mainly descriptive. However, we tried to find associations among study parameters.

  1. The causalities between GMA and dietary intakes, substrate utilization, and energy expenditure in obese subjects were not well defined and supported by a clear hypothesis.

Response: sorry for this oversight. However, throughout the introduction, we tried to describe the available knowledge about the impact of various dietary intakes on the GMA. In lines 102 to 106, our hypothesis was clarified.

  1. In this study, the significant correlations between different parameters seemed to be observed randomly observed in a specific group. There was no consistent correlation of a particular parameter established with the progression of the severity of obesity,detract significantly from the conceptual novelty of the findings.

Response: The selection of the parameters which were tested by correlation coefficients was based on the rationale and the objective of this study. The absence of a consistent correlation of a particular parameter with the severity of obesity may be due to the subdivision of the participants with obesity into obese and morbidly obese. In figure 1 we tested the correlation if we used only 2 groups (Obesity and No obesity) which may solve this issue (if you agree with this change). Also, metabolic and hormonal changes in morbid obesity may disturb the associations in severe forms of obesity.

Figure 1: Crude correlation tables of study groups as two groups instead of 4 groups.

Round 2

Reviewer 2 Report

No further question.

Author Response

Thanks. Really appreciate the support